# The Extracts of *Artemisia absinthium* L. Suppress the Growth of Hepatocellular Carcinoma Cells through Induction of Apoptosis via Endoplasmic Reticulum Stress and Mitochondrial-Dependent Pathway

**DOI:** 10.3390/molecules24050913

**Published:** 2019-03-05

**Authors:** Xianxian Wei, Lijie Xia, Dilinigeer Ziyayiding, Qiuyan Chen, Runqing Liu, Xiaoyu Xu, Jinyao Li

**Affiliations:** Xinjiang Key Laboratory of Biological Resources and Genetic Engineering, College of Life Science and Technology, Xinjiang University, Urumqi, Xinjiang 830046, China; 15099141611@189.cn (X.W.); xialijie1219@163.com (L.X.); dilnigar9696@sina.com (D.Z.); m15276567620_1@163.com (Q.C.); Lucyducy@163.com (R.L.); 15276654427@139.com (X.X.)

**Keywords:** *Artemisia absinthium*, apoptosis, endoplasmic reticulum stress, mitochondrial-dependent pathway

## Abstract

*Artemisia absinthium* L. has pharmaceutical and medicinal effects such as antimicrobial, antiparasitic, hepatoprotective, and antioxidant activities. Here, we prepared *A. absinthium* ethanol extract (AAEE) and its subfractions including petroleum ether (AAEE-Pe) and ethyl acetate (AAEE-Ea) and investigated their antitumor effect on human hepatoma BEL-7404 cells and mouse hepatoma H22 cells. The cell viability of hepatoma cells was measured by 3-(4,5-dimethylthiazol-2-yl)-2,5-diphenyltetrazolium bromide (MTT) assay. The apoptosis, cell cycle, mitochondrial membrane potential (Δψm), and reactive oxygen species (ROS) were analyzed by flow cytometry. The levels of proteins in the cell cycle and apoptotic pathways were detected by Western blot. AAEE, AAEE-Pe, and AAEE-Ea exhibited potent cytotoxicity for both BEL-7404 cells and H22 cells through the induction of cell apoptosis and cell cycle arrest. Moreover, AAEE, AAEE-Pe, and AAEE-Ea significantly reduced Δψm, increased the release of cytochrome c, and promoted the cleavage of caspase-3, caspase-9, and poly(ADP-ribose) polymerase (PARP) in BEL-7404 and H22 cells. AAEE, AAEE-Pe, and AAEE-Ea significantly upregulated the levels of ROS and C/EBP-homologous protein (CHOP). Further, AAEE, AAEE-Pe, and AAEE-Ea significantly inhibited tumor growth in the H22 tumor mouse model and improved the survival of tumor mice without side effects. These results suggest that AAEE, AAEE-Pe, and AAEE-Ea inhibited the growth of hepatoma cells through induction of apoptosis, which might be mediated by the endoplasmic reticulum stress and mitochondrial-dependent pathway.

## 1. Introduction

Hepatocellular carcinoma (HCC) is one of the most common malignant tumors and was the sixth most commonly diagnosed cancer and the fourth leading cause of cancer death worldwide in 2018, with about 841,000 new cases and 782,000 deaths annually [1,2]. The main risk factors for HCC are chronic infection with hepatitis B virus (HBV) or hepatitis C virus (HCV), aflatoxin-contaminated foodstuffs, heavy alcohol intake, obesity, smoking, and type 2 diabetes [3]. Currently, therapeutic options for the treatment of HCC include liver resection, transplantation, palliative intra-arterial therapies, immunotherapy strategies, and so on [4,5]. However, the prognosis of most patients with HCC is poor [6]. For HCC treatment, the main drugs, including oxaliplatin and sorafenib, remain unsatisfactory because of their side effects and multidrug resistance [7,8]. Therefore, it is urgent to develop novel therapeutic agents to treat HCC.

*Artemisia absinthium* L. belongs to the Asteraceae family and is commonly known as wormwood. The chemical components of *A. absinthium* include sesquiterpene lactone, sesquiterpene lactone-pinene, β-thujone, α-thujone, sabinyl acetate, and β-thujone [9]. *A. absinthium* has pharmaceutical and medicinal effects such as antimicrobial [9], insecticidal [10], antiparasitic [11], antitumor [12], antipyretic [13], hepatoprotective [14,15], and antioxidant activities [16,17]. In the present study, *A. absinthium* ethanol extract (AAEE) and its subfractions including petroleum ether (AAEE-Pe) and ethyl acetate (AAEE-Ea) were prepared, and their antitumor effects on HCC were investigated both in vitro and in vivo. AAEE, AAEE-Pe, and AAEE-Ea selectively inhibited the growth of hepatoma cells both in vitro and in vivo without cytotoxic effects on normal hepatic cells. Moreover, these extracts could arrest the cell cycle at the G2/M phase and induce apoptosis through endoplasmic reticulum (ER) stress and the mitochondrial-dependent pathway in human hepatoma BEL-7404 cells and mouse hepatoma H22 cells, and they might be used as safe and effective agents for the treatment of HCC.

## 2. Results

### 2.1. AAEE, AAEE-Pe, and AAEE-Ea Suppress the Growth of BEL-7404 and H22 Cells In Vitro

To investigate the anti-proliferative effects of AAEE, AAEE-Pe, and AAEE-Ea, BEL-7404 and H22 cells were treated with 25, 75, and 150 μg/mL of AAEE, AAEE-Pe, and AAEE-Ea. After 24 h, the morphology of BEL-7404 and H22 cells was observed with an inverted microscope. Compared to untreated cells, BEL-7404 and H22 cells treated with AAEE, AAEE-Pe, and AAEE-Ea became shrunk and round, and cell numbers were reduced in a dose-dependent manner (Figure 1A). Cell viability was detected by MTT assay after treatment for 24, 48, and 72 h. The viability of BEL-7404 and H22 cells was dose- and time-dependently decreased after treatment with AAEE, AAEE-Pe, or AAEE-Ea (Figure 1B). The IC_50_ (50% inhibitory concentration) values of AAEE, AAEE-Pe, and AAEE-Ea for BEL-7404 and H22 cells at 24, 48, and 72 h are shown in Table 1. The IC_50_ values of H22 cells followed the order AAEE-Pe ≤ AAEE < AAEE-Ea. The IC_50_ values of BEL-7404 cells followed the order AAEE ≤ AAEE-Pe < AAEE-Ea. The effect of AAEE, AAEE-Pe, and AAEE-Ea was also detected on normal liver cells NCTC1469. AAEE and AAEE-Pe showed some cytotoxicity on NCTC1469 cells, but it was much lower than that of BEL-7404 and H22 cells. AAEE-Ea has no cytotoxicity on NCTC1469 cells (Figure 1C). These results suggest that AAEE, AAEE-Pe, and AAEE-Ea selectively inhibited the growth of hepatoma cells in vitro.

### 2.2. AAEE, AAEE-Pe, and AAEE-Ea Induce Apoptosis in BEL-7404 and H22 Cells

To study whether AAEE, AAEE-Pe, and AAEE-Ea induce apoptosis, BEL-7404 and H22 cells were stained with Annexin V and propidium iodide (PI) after treatment and analyzed by flow cytometry. The frequencies of apoptotic BEL-7404 and B22 cells were significantly increased by each of AAEE, AAEE-Pe, and AAEE-Ea in a dose-dependent manner (Figure 2A,B). AAEE, AAEE-Pe, and AAEE-Ea did not induce the necrosis of BEL-7404 cells but significantly induced necrosis of H22 cells (Figure 2A,B). The pro- and anti-apoptotic members of the B-cell lymphoma-2 (BCL-2) protein family serve important roles in the regulation of cell apoptosis. After treatment with AAEE, AAEE-Pe, and AAEE-Ea, the levels of pro-apoptotic Bax and anti-apoptotic Bcl-2 in BEL-7404 and H22 cells were upregulated and downregulated, respectively. The ratios of Bax/Bcl2 significantly increased upon AAEE, AAEE-Pe, and AAEE-Ea treatment (Figure 2C). The results indicate that AAEE, AAEE-Pe, and AAEE-Ea inhibited the growth of BEL-7404 and H22 cells through the induction of apoptosis.

### 2.3. AAEE, AAEE-Pe, and AAEE-Ea Induce Cell Cycle Arrest in BEL-7404 and H22 Cells

The morphological characteristics of apoptosis include chromatin condensation and DNA fragmentation. After treatment with AAEE, AAEE-Pe, and AAEE-Ea, BEL-7404 and H22 cells were stained with Hoechst 33258 and observed by inverted fluorescence microscopy. The nuclei of untreated cells showed homogeneous staining, while the nuclei of cells treated with AAEE, AAEE-Pe, and AAEE-Ea showed condensed chromatin (Figure 3A), suggesting that AAEE, AAEE-Pe, and AAEE-Ea induce apoptosis in BEL-7404 and H22 cells.

Next, the distribution of the cell cycle in BEL-7404 and H22 cells was detected by PI staining after treatment with AAEE, AAEE-Pe, and AAEE-Ea for 24 h. We observed that cells in the G2/M phase were significantly increased and cells in the S phases were significantly decreased upon AAEE, AAEE-Pe, and AAEE-Ea treatment (Figure 3B), indicating that AAEE, AAEE-Pe, and AAEE-Ea arrested the cell cycle of BEL-7404 and H22 cells at the G2/M phase. Consistently, AAEE, AAEE-Pe, and AAEE-Ea reduced the expression of cyclin B1 in BEL-7404 and H22 cells (Figure 3C). The results suggest that AAEE, AAEE-Pe, and AAEE-Ea induced apoptosis and cell cycle arrest in hepatoma cells.

### 2.4. AAEE, AAEE-Pe, and AAEE-Ea Reduce Mitochondrial Membrane Potential (Δψm)

Mitochondrial membrane integrity is strictly regulated by the pro- and anti-apoptotic members of the BCL-2 protein family. After treatment with AAEE, AAEE-Pe, and AAEE-Ea for 24 h, the Δψm was measured by flow cytometry using JC-1 as a fluorescent dye. As shown in Figure 4A, AAEE, AAEE-Pe, and AAEE-Ea significantly reduced the Δψm values of BEL-7404 and H22 cells in a dose-dependent manner. Consistently, AAEE, AAEE-Pe, and AAEE-Ea promoted the release of cytochrome c in the cytosol of BEL-7404 and H22 cells. The activation of caspases is generally considered to be a key hallmark of apoptosis. We also observed that the levels of cleaved caspase-3 and caspase-9 were increased upon AAEE, AAEE-Pe, and AAEE-Ea treatment, which promoted the cleavage of poly(ADP-ribose) polymerase (PARP) in both BEL-7404 and H22 cells (Figure 4B). The results indicate that AAEE, AAEE-Pe, and AAEE-Ea induced apoptosis in hepatoma cells through the mitochondria-dependent pathway.

### 2.5. AAEE, AAEE-Pe, and AAEE-Ea Promote Reactive Oxygen Species (ROS) Generation and ER Stress

To examine the effects of AAEE, AAEE-Pe, and AAEE-Ea on oxidative stress, ROS generation was detected by flow cytometry at the indicated time points. After treatment with AAEE, AAEE-Pe, and AAEE-Ea, ROS levels were significantly increased at 6 h, reached a peak at 12 h, and then were maintained to 24 h in BEL-7404 cells (Figure 5A). Similarly, AAEE, AAEE-Pe, and AAEE-Ea dose-dependently increased ROS levels in H22 cells after treatment for 24 h (Figure 5B).

ER stress exacerbates mitochondrial dysfunction by activating caspase-9 and increasing the release of cytochrome c [18]. CHOP, DNA damage inducible gene 153 (GADD153), is the main apoptotic factor activated by ER stress, and its overexpression promotes apoptosis in cancer [19]. After treatment with AAEE, AAEE-Pe, and AAEE-Ea for 24 h, the levels of CHOP were detected by Western blot. The results showed that AAEE, AAEE-Pe, and AAEE-Ea significantly increased the levels of CHOP in BEL-7404 and H22 cells (Figure 5C), suggesting that ER stress might be involved in the induction of apoptosis by AAEE, AAEE-Pe, and AAEE-Ea.

### 2.6. AAEE, AAEE-Pe, and AAEE-Ea Inhibit the Growth of H22 Cells In Vivo

To further confirm the inhibitory effect of AAEE, AAEE-Pe, and AAEE-Ea on tumor growth in vivo, an H22 tumor mouse model was established in Kunming mice. Tumor mice were treated with different doses of AAEE, AAEE-Pe, and AAEE-Ea after 3 days of H22 cell injection. Cisplatin was used as a positive control and DMSO was used as a solvent control. We observed that cisplatin significantly reduced the weight of mice, but AAEE, AAEE-Pe, and AAEE-Ea did not affect the weight of mice. The tumor growth was significantly inhibited in the cisplatin, 100 mg/kg AAEE-Pe, and 100 and 200 mg/kg AAEE-Ea groups. AAEE at a 200 mg/kg dose also suppressed tumor growth to a certain degree (Figure 6A). At the end of the tumor study, the survival rates of tumor mice in each group were calculated. All mice were dead in the control (7 out of 7) and DMSO (7 out of 7) groups. The survival rates were 37.5%, 25%, 37.5%, 50%, and 87.5% in the cisplatin (3 out of 8), 200 mg/kg AAEE (2 out of 8), 100 mg/kg AAEE-Pe (3 out of 8), and 100 (4 out of 8) and 200 mg/kg (7 out of 8) AAEE-Ea groups, respectively (Figure 6B). These data suggest that AAEE, AAEE-Pe, and AAEE-Ea effectively inhibited the growth of H22 cells in vivo and improved the survival of tumor mice without obvious toxicity.

## 3. Discussion

Chinese herbal medicine (CHM) has a long history of use in treating cancers and provides potential antitumor remedies. *A. absinthium*, a kind of CHM, has been used as an antipyretic, antiseptic, and anti-parasitic agent for the treatment of chronic fevers and inflammation of the liver [20]. In this study, we prepared AAEE, AAEE-Pe, and AAEE-Ea and investigated their antitumor effects on hepatoma cells. We found that AAEE, AAEE-Pe, and AAEE-Ea significantly suppressed the growth of BEL-7404 and H22 cells, induced apoptosis and cell cycle arrest, reduced Δψm, increased the release of cytochrome c, activated caspases, and promoted ROS production and ER stress.

It has been reported that a number of components of CHM can inhibit the growth of tumor cells both in vitro and in vivo, such as polysaccharides, flavones, terpenoids, and phenols [21,22,23,24,25,26]. The components of polysaccharides, flavones, and triterpenes in AAEE, AAEE-Pe, and AAEE-Ea were quantified. Although the three extracts contained different concentrations of polysaccharides, flavones, and triterpenes, they had similar antitumor effects. The results suggest that polysaccharides, flavones, and triterpenes might not be the major antitumor components in AAEE, AAEE-Pe, and AAEE-Ea. We will further identify the major antitumor components in the extracts of *A. absinthium* in future study.

The BCL-2 protein family strictly controls the apoptosis of cells [27,28]. An imbalance of proteins in the BCL-2 family triggers the intrinsic apoptosis pathway that increases mitochondrial permeability and the release of cytochrome c and activates caspase-9/caspase-3 [29]. We found that the levels of Bax/Bcl2 significantly increased in BEL-7404 and H22 cells after AAEE, AAEE-Pe, and AAEE-Ea treatment, which might cause the reduction of Δψm and the release of cytochrome c to activate caspases and apoptosis. Similarly, Shafi et al. [30] reported that the methanol extract of *A. absinthium* induced the apoptosis of human breast cancer cells through the modulation of BCL-2 family proteins.

Accumulating evidence points to the role of ER stress in the induction of apoptosis in various cancer cells [31,32,33]. ER stress exacerbates mitochondrial dysfunction by activating caspase-9 and increasing the release of cytochrome c [34,35]. Our results showed that AAEE, AAEE-Pe, and AAEE-Ea treatment significantly increased the expression of the ER stress-related protein CHOP, indicating that AAEE, AAEE-Pe, and AAEE-Ea might induce apoptosis of BEL-7404 and H22 cells through ER stress and the mitochondria-dependent pathway.

We observed that the ROS levels in hepatoma cells treated with AAEE and AAEE-Pe were lower than that in untreated cells at the beginning of 3 h, although they were significantly upregulated after 6 h, suggesting that the extracts might have antioxidant activities. This is similar to results found in other studies [16,17]. The increased ROS might be due to the reduction of Δψm induced by AAEE and AAEE-Pe treatment.

Finally, the antitumor and side effects of AAEE, AAEE-Pe, and AAEE-Ea were evaluated in an H22 tumor mouse model. All three extracts did not affect the weight of mouse but cisplatin significantly reduced the weight of mouse, suggesting that AAEE, AAEE-Pe, and AAEE-Ea might have no side effects in vivo. Moreover, AAEE-Pe and AAEE-Ea significantly inhibited tumor growth, which was similar with cisplatin. AAEE, AAEE-Pe, and AAEE-Ea further improved the survival of H22 tumor mice, and the high dose of AAEE-Ea showed the highest survival rate.

In conclusion, AAEE, AAEE-Pe, and AAEE-Ea inhibited the growth of hepatoma cells through the induction of apoptosis that might be mediated by ER stress and the mitochondria-dependent pathway. AAEE, AAEE-Pe, and AAEE-Ea suppressed H22 tumor growth and improved the survival of H22 tumor mice without obvious side effects, indicating that the three extracts might be used to develop safe and effective antitumor agents.

## 4. Materials and Methods

### 4.1. Preparation of AAEE, AAEE-Pe, and AAEE-Ea

The AAEE, AAEE-Pe, and AAEE-Ea were prepared according to the following procedure. Briefly, the powder was made using the aerial parts of *A. absinthium* including stems, leaves, flowers, and seeds (Alikang Uygur medicine technology co., Ltd., Urumqi, Xinjiang, China) and extracted overnight using 10 volumes of distilled water at 4 °C. After centrifugation at 8000 rpm for 20 min, the pellet was collected and extracted with 10 volumes of distilled water for 2 h at 60 °C. After centrifugation, the pellet was extracted using 10 volumes of 85% ethanol at 60 °C three times (2 h/time). The supernatant was collected after filtration and concentrated by a rotary evaporator to obtain the extractum. Some extractum was dried by a vacuum freeze-dryer to obtain AAEE. The remaining extractum was dissolved in distilled water and extracted by an equal volume of petroleum ether eight times, then the upper layer was collected and dried by a vacuum freeze–dryer to obtain AAEE-Pe. The bottom layer was extracted by an equal volume of ethyl acetate eight times, then the supernatant was collected and dried by a vacuum freeze–dryer to obtain AAEE-Ea. The AAEE, AAEE-Pe, and AAEE-Ea were dissolved in dimethyl sulfoxide (DMSO) (Sigma, St. Louis, MO, USA) and filtered with a 0.22 μm filter. The contents of flavonoids, terpenoids, and polysaccharides were determined by AlCl_3_–KA_C_, vanillin–glacial acetic acid, and anthrone–sulfuric acid colorimetry, respectively, which were shown in Table 2.

### 4.2. Cell Culture

The NCTC1469, H22, and BEL-7404 cells were obtained from the Xinjiang Key Laboratory of Biological Resources and Genetic Engineering, Xinjiang University (Urumqi, Xinjiang, China) and cultured in Roswell Park Memorial Institute (RPMI) 1640 medium (Gibco, Thermo Fisher Scientific, Waltham, MA, USA) supplemented with 10% fetal bovine serum (MRC, Changzhou, China) and 1% l-glutamine (100 mM), 100 U/mL penicillin, and 100 μg/mL streptomycin (MRC, Changzhou, China) at 37 °C in humidified air with 5% CO_2_.

### 4.3. Cell Viability Assay

The proliferation of NCTC1469, H22, and BEL-7404 cells was analyzed by 3-(4,5-dimethylthiazol-2-yl)-2,5-diphenyltetrazolium bromide (MTT) (Sigma, Louis, MO, USA) assay. Briefly, cells (5000 cells/well) were seeded in 96-well plates and treated with various doses of AAEE, AAEE-Pe, and AAEE-Ea for 24 h, 48 h, or 72 h. DMSO (0.3%) was used as a solvent control and cisplatin (35 μg/mL) was used as a positive control. The supernatant was discarded after centrifugation at 1200 rpm for 5 min and 100 μL of MTT solution (0.5 mg/mL in PBS) was added to each well and incubated at 37 °C for 3 h. The formed formazan crystals were dissolved in 200 μL DMSO. The OD_490_ values were measured by a 96-well microplate reader (Bio-Rad Laboratories, Hercules, CA, USA). The relative cell viability was calculated according to the formula Cell viability (%) = (OD_treated_/OD_untreated_) × 100%. This experiment was conducted three times independently.

### 4.4. Observation of Cell Morphology

H22 and BEL-7404 cells were seeded in 96-well plates and were treated with different concentrations of AAEE, AAEE-Pe, and AAEE-Ea for 24 h. After treatment, the morphology of H22 and BEL-7404 cells was observed by inverted fluorescence microscope (Nikon Eclipse Ti-E, Tokyo, Japan).

### 4.5. Analysis of Apoptosis

H22 and BEL-7404 cells were treated with different concentrations of AAEE, AAEE-Pe, and AAEE-Ea for 24 h, and then stained with an Annexin V-FITC/propidium iodide (PI) Apoptosis Detection Kit (YEASEN, Shanghai, China) according to the manufacturer’s instructions. Cisplatin and DMSO were used as positive and negative controls, respectively. Samples were analyzed by flow cytometry (BD FACSCalibur, San Jose, CA, USA). This experiment was conducted three times independently.

### 4.6. Hoechst 33258 Staining

H22 and BEL-7404 cells were seeded in 6-well plates at the concentration of 1 × 10^5^ cells/well in 2 mL medium. After 60%~70% confluence, the cells were treated with AAEE, AAEE-Pe, and AAEE-Ea for 24 h. The cells were collected and fixed with 4% ice-cold Paraformaldehyde at 4 °C for 10 min. After washing with PBS, cells were stained with Hoechst 33258 (Beyotime, Shanghai, China) at 4 °C for 10 min. Samples were observed using an inverted fluorescence microscope.

### 4.7. Analysis of the Cell Cycle

H22 and BEL-7404 cells were inoculated in 60 mm culture dishes and treated with different concentrations of AAEE, AAEE-Pe, and AAEE-Ea for 24 h. All cells were collected and washed twice with PBS, then fixed in 70% ice-cold ethanol overnight at 4 °C. After washing twice with PBS, cells were re-suspended in 250 μL propidium iodide/RNase staining buffer (BD Biosciences, San Jose, CA, USA). After 10 min at room temperature, samples were collected by flow cytometry and the cell cycle distribution was analyzed using ModFit LT 3.0 software (BD FACS Calibur, San Jose, AC, USA). This experiment was conducted three times independently.

### 4.8. Analysis of Δψm

H22 and BEL-7404 cells were treated with different concentrations of AAEE, AAEE-Pe, and AAEE-Ea for 24 h and then stained with the membrane-permeable JC-1 dye (Beyotime, Shanghai, China) for 30 min at 37 °C. Samples were analyzed by flow cytometry. This experiment was conducted three times independently.

### 4.9. Analysis of ROS

BEL-7404 cells were treated with AAEE, AAEE-Pe, and AAEE-Ea for 2, 4, 6, 12, and 24 h. H22 cells were treated with AAEE, AAEE-Pe, and AAEE-Ea for 24 h. Cells were stained by 10 mM of fluorescent probe 2′,7′-dichlorodihydrofluorescein diacetate (DCFH-DA) (Beyotime, Shanghai, China) for 20 min at 37 °C. After washing three times with ice-cold PBS, samples were analyzed by flow cytometry. This experiment was conducted two times independently.

### 4.10. Western Blot

The antibodies against caspase-9, Bax and Bcl-2, and anti-mouse IgG-HRP and anti-rabbit IgG-HRP were purchased from BBI Life Sciences (Shanghai, China). The antibodies against caspase-3, PARP, cytochrome c, and β-actin were obtained from Cell Signaling Technology (Danvers, MA, USA). The antibodies against CHOP and CyclinB1 were bought from Beyotime (Shanghai, China).

H22 and BEL-7404 cells were treated with AAEE, AAEE-Pe, and AAEE-Ea for 24 h. After washing twice with PBS, cell lysates were prepared with RIPA Lysis Buffer (Beijing ComWin Biotech Co., Ltd., Beijing, China) and protein concentrations were detected using a BCA Kit (Thermo Fisher Scientific, Waltham, MA, USA) according to the manufacturer’s instructions. Proteins were separated on 12% sodium dodecyl sulfate polyacrylamide gel electrophoresis (SDS-PAGE) and transferred to a polyvinylidene difluoride (PVDF) membrane. After incubation with primary and secondary antibodies, target proteins were detected by chemiluminescence (Beyotime, Shaghai, China). Signals were quantified using ImageJ digitizing software (ImageJ 1.50, National Institutes of Health, Bethesda, MD, USA). This experiment was conducted three times independently.

### 4.11. Animals and Ethics Statement

Six- to eight-week-old male Kunming mice were purchased from Animal Laboratory Center, Xinjiang Medical University (Urumqi, Xinjiang, China). Mice were kept in a standard temperature-controlled, light-cycled animal facility at Xinjiang University. All animal experiments were approved by the Committee on the Ethics of Animal Experiments of Xinjiang Key Laboratory of Biological Resources and Genetic Engineering (BRGE-AE001) and performed under the guidelines of the Animal Care and Use Committee of College of Life Science and Technology, Xinjiang University.

### 4.12. Tumor Mouse Study

For establishment of a tumor mouse model, male Kunming mice were injected with 1 × 10^6^ H22 cells in 100 μL PBS subcutaneously. After 3 days, mice were randomly divided into seven groups (7 mice/group for Control and DMSO, 8 mice/group for the other five groups). The solvent control group intraperitoneally received 0.1 mL DMSO daily. The positive group was intraperitoneally injected with 5 mg/kg cisplatin at intervals of five days. The experimental groups were intraperitoneally injected with 200 mg/kg AAEE, 100 mg/kg AAEE-Pe, or 100 mg/kg or 200 mg/kg AAEE-Ea in 0.1 mL DMSO every two days. Tumor sizes were measured using calipers and tumor volume was calculated according to the formula tumor volume (mm^3^) = (length × width^2^)/2.

### 4.13. Statistical Analysis

The data are expressed as mean ± standard error of the mean (SEM). Statistical significance was analyzed using one-way analysis of variance (ANOVA) by Tukey’s Multiple Comparison Test. *p* < 0.05 was considered statistically significant.

## Figures and Tables

**Figure 1 molecules-24-00913-f001:**
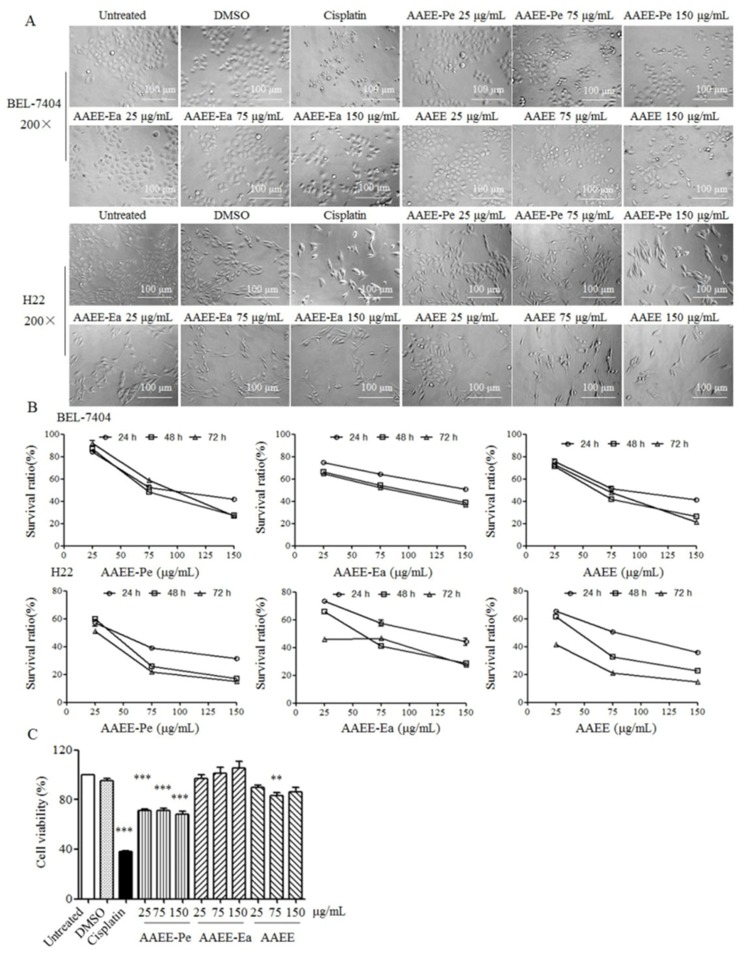
The effect of *A. absinthium* ethanol extract (AAEE) and its petroleum ether (AAEE-Pe) and ethyl acetate (AAEE-Ea) subfractions on the growth of BEL-7404, H22, and NCTC1469 cells. Cells were treated with different concentrations of AAEE, AAEE-Pe, and AAEE-Ea. (**A**) After 24 h, the morphology of BEL-7404 and H22 cells was observed by inverted microscope. (**B**) After 24, 48, and 72 h, the viability of BEL-7404 and H22 cells was detected by MTT assay. (**C**) After 24 h, the viability of NCTC1469 cells was detected by MTT assay. ** *p* < 0.01, *** *p* < 0.001 compared to Untreated.

**Figure 2 molecules-24-00913-f002:**
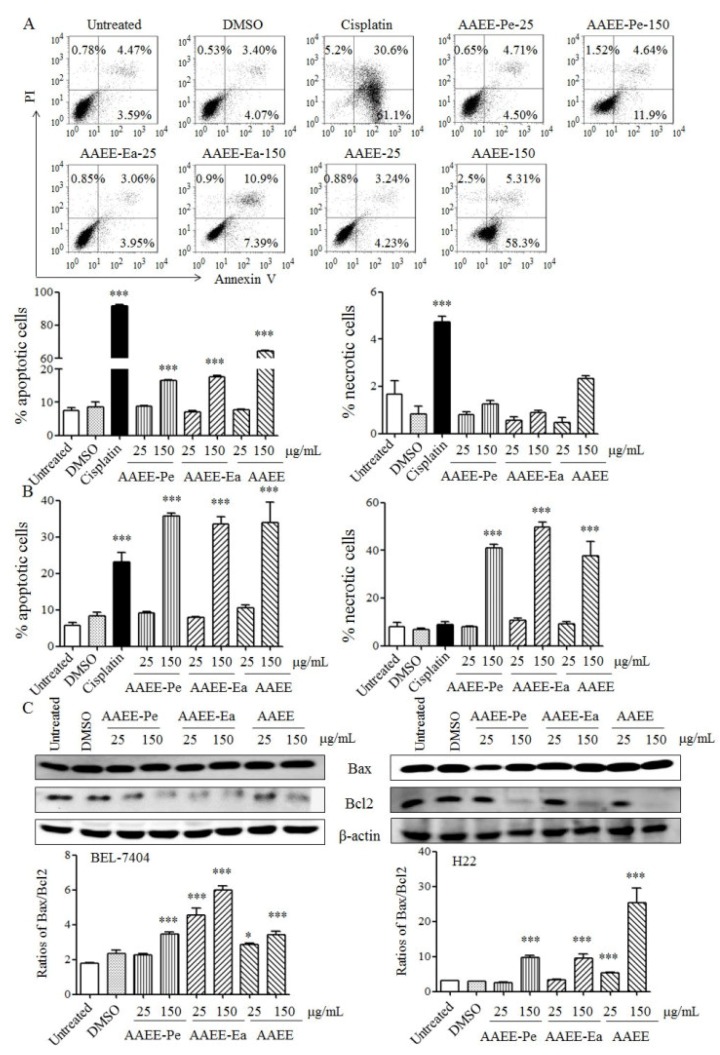
AAEE, AAEE-Pe, and AAEE-Ea induced apoptosis in BEL-7404 and H22 cells. Cells were treated with different concentrations of AAEE, AAEE-Pe, and AAEE-Ea for 24 h. After staining with Annexin V and PI, BEL-7404 (**A**) and H22 (**B**) cells were analyzed by flow cytometry. (**C**) Proteins were isolated, and the levels of Bax and Bcl-2 were analyzed by Western blot. * *p* < 0.05; *** *p* < 0.001 compared to Untreated.

**Figure 3 molecules-24-00913-f003:**
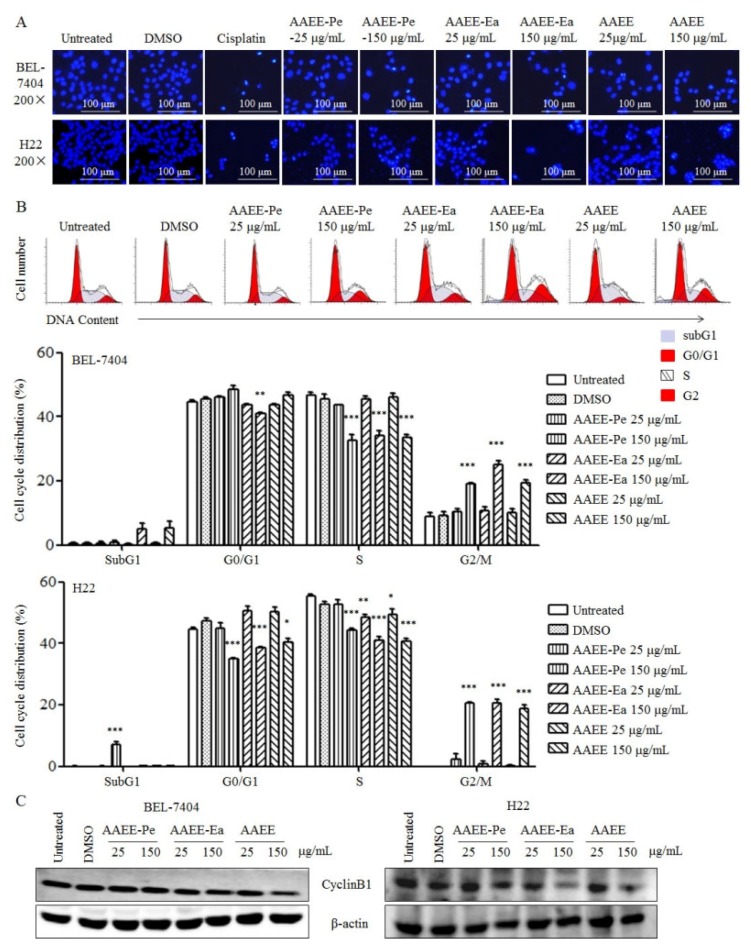
AAEE, AAEE-Pe, and AAEE-Ea induced cell cycle arrest in BEL-7404 and H22 cells. BEL-7404 and H22 cells were treated with different concentrations of AAEE, AAEE-Pe, and AAEE-Ea for 24 h. (**A**) BEL-7404 and H22 cells were stained with Hoechst 33258 and observed by inverted fluorescence microscope. (**B**) DNA contents in BEL-7404 cells were analyzed by flow cytometry and are shown in the upper panels. The summaries of the cell cycle distributions in BEL-7404 and H22 cells are shown in the middle and bottom panels, respectively. (**C**) Expression of Cyclin B1 was analyzed by Western blot. * *p* < 0.05; ** *p* < 0.01; *** *p* < 0.001 compared to Untreated.

**Figure 4 molecules-24-00913-f004:**
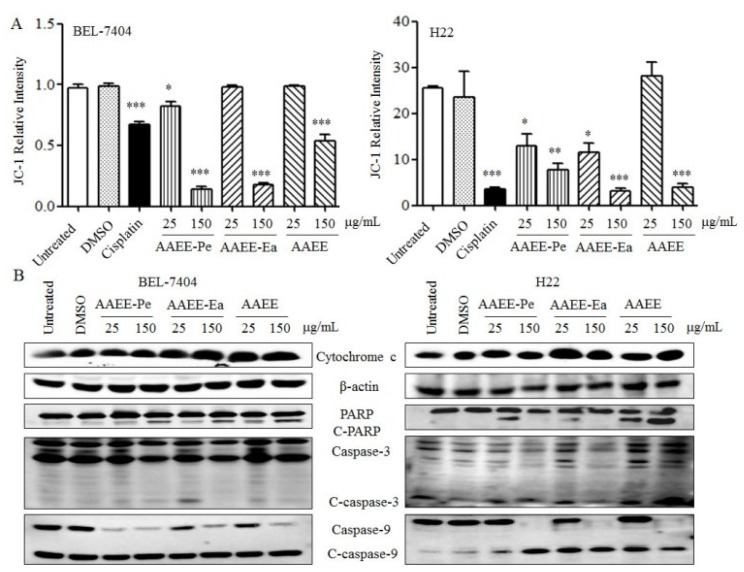
The mitochondria-dependent apoptosis induced by AAEE, AAEE-Pe, and AAEE-Ea. BEL-7404 and H22 cells were treated with different concentrations of AAEE, AAEE-Pe and AAEE-Ea for 24 h. (**A**) Cells were stained with JC-1 dye and analyzed by flow cytometry. (**B**) Total protein was isolated to detect the release of cytochrome c and the levels of poly(ADP-ribose) polymerase (PARP), cleaved-PARP (C-PARP), cleaved-caspase-3, caspase-3, cleaved-caspase-9, and caspase-9 by Western blot. * *p* < 0.05; ** *p* < 0.01; *** *p* < 0.001 compared to Untreated.

**Figure 5 molecules-24-00913-f005:**
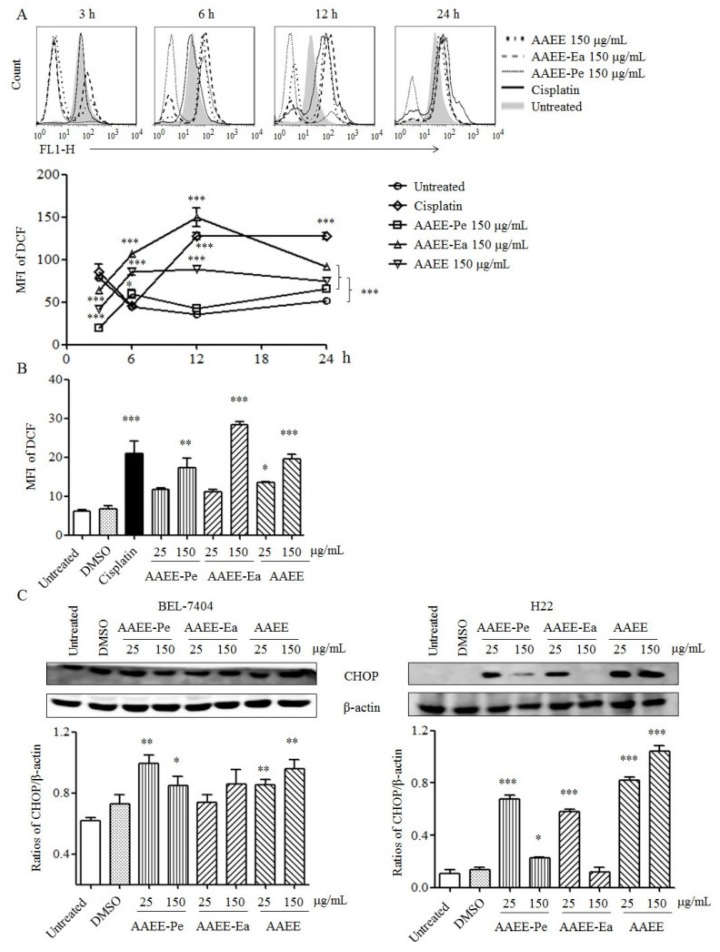
AAEE, AAEE-Pe, and AAEE-Ea induced ROS generation and ER stress. (**A**) BEL-7404 cells were treated with 150 μg/mL of AAEE, AAEE-Pe, and AAEE-Ea and the levels of ROS were analyzed by flow cytometry at indicated time points. (**B**) H22 cells were treated with different concentrations of AAEE, AAEE-Pe, and AAEE-Ea for 24 h and the levels of ROS were analyzed by flow cytometry. (**C**) BEL-7404 and H22 cells were treated with different concentrations of AAEE, AAEE-Pe, and AAEE-Ea for 24 h. Cell lysates were used to analyze the levels of CHOP by Western blot. * *p* < 0.05; ** *p* < 0.01; *** *p* < 0.001 compared to Untreated.

**Figure 6 molecules-24-00913-f006:**
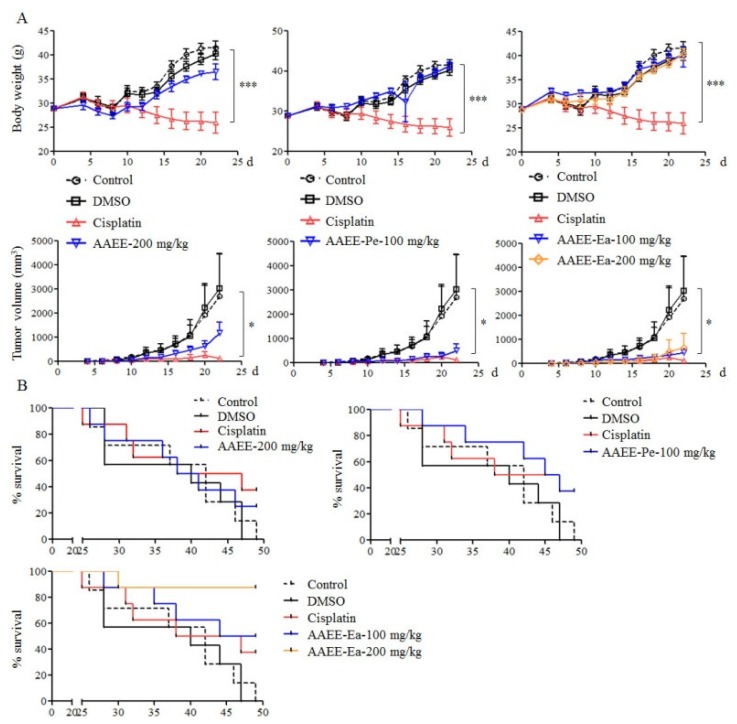
AAEE, AAEE-Pe, and AAEE-Ea suppressed tumor growth in vivo. A tumor mouse model was established by injection of H22 cells. After 3 days, tumor mice were treated with DMSO, cisplatin, AAEE, AAEE-Pe, and AAEE-Ea. Body weight of mice, tumor sizes (**A**), and survival rates (**B**) were monitored at the indicated time points. * *p* < 0.05; *** *p* < 0.001 compared to control.

**Table 1 molecules-24-00913-t001:** IC_50_ values of AAEE, AAEE-Pe, and AAEE-Ea for BEL-7404 and H22 cells.

		IC_50_		
		24 h	48 h	72 h
H22 cells	AAEE	59.16	33.40	14.71
	AAEE-Pe	32.91	29.64	23.38
	AAEE-Ea	97.76	43.57	25.69
BEL-7404 cells	AAEE	89.86	56.86	60.39
	AAEE-Pe	98.71	75.88	89.51
	AAEE-Ea	171.7	82.04	71.97

**Table 2 molecules-24-00913-t002:** The contents of polysaccharides, flavonoids, and triterpenoids in AAEE, AAEE-Pe, and AAEE-Ea.

	AAEE	AAEE-Pe	AAEE-Ea
Polysaccharides	31.15%	0.68%	9.62%
Flavonoids	16.89%	10.26%	24.97%
Triterpenes	22.57%	31.59%	29.89%

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
