# Peer review of "The Extracts of Artemisia absinthium L. Suppress the Growth of Hepatocellular Carcinoma Cells through Induction of Apoptosis via Endoplasmic Reticulum Stress and Mitochondrial-Dependent Pathway"

_molecules, 2019, doi:10.3390/molecules24050913_

Reviewer 1 Report

This paper by Wei, et al, demonstrated that the in vivo and vitro antitumor effects of Artemisia Absinthium extract on Hepatocellular carcinoma. The general purpose of this study is clear, and the study appears to be of interest. However, I recommend you to add more information and/or data about ER stress-dependent pathway. Overall I recommend this paper to be published in Molecules after addressing the following comments.

Major Point
Results & Conclusion

The authors clearly show that Artemisia Absinthium extract increased the expression of CHOP in human hepatoma BEL-7404 cells and mouse hepatoma H22 cells. From the results obtained in these results, the authors concluded that the Artemisia Absinthium extract induced apoptosis in mediated by endoplasmic reticulum stress pathway. However, I feel that this conclusion overinterpret the evidence. I recommend you to add more data about ER stress-related protein and/or gene expression levels, such as GRP78, PERK, ATF6, IRE1α and Caspase-12.The authors should explain this point in the revised manuscript. 

Figure

The resolution of all Figure is poor. Especially, the quality of photomicrograph in Fig. 1 is poor. The author should replace the original image to a better image in the revised manuscript.

Method

Please indicate the number of each experiments. Especially, you should describe the total number of animals used in each experiment.

Discussion

In the Introduction section, the authors described A. absinthium has pharmaceutical and 48 medicinal effects such as antimicrobial [9], insecticidal [10], antiparasitic [11], antitumor [12], 49 antipyretic [13], hepatoprotective [14,15] and antioxidant activities [16,17]. As you described in the introduction section, A. absinthium has an antioxidant activities. In this manuscript, however, you showed that these extract promoted ROS production in BEL-7404 cells and H22 cells. The authors should address this seemingly paradoxical consequences in the revised manuscript.

Mainor point

What kind of post hoc tests are you using in this study?

 Please add the representative bands of Internal Standard in fig2C.

Author Response

Major Point
Results & Conclusion

The authors clearly show that Artemisia Absinthium extract increased the expression of CHOP in human hepatoma BEL-7404 cells and mouse hepatoma H22 cells. From the results obtained in these results, the authors concluded that the Artemisia Absinthium extract induced apoptosis in mediated by endoplasmic reticulum stress pathway. However, I feel that this conclusion overinterpret the evidence. I recommend you to add more data about ER stress-related protein and/or gene expression levels, such as GRP78, PERK, ATF6, IRE1α and Caspase-12.The authors should explain this point in the revised manuscript. 

Answer: We appreciate the reviewer’s suggestion. The upregulation of CHOP expression is a hallmark of endoplasmic reticulum stress but it is involved in different pathways such as IRE-1-CHOP, ATF6-CHOP, and PERK-ATF4-eIF2a-CHOP. The extracts contained various components that might induce ER stress through different pathways. Therefore, we just detected the expression of CHOP. We have changed the description in the revised manuscript. In the future study, we will purify the active components and investigate ER stress pathways.

 Method

Please indicate the number of each experiment. Especially, you should describe the total number of animals used in each experiment.

Answer: We are sorry for the missing information. We have added the information in the revised manuscript.

 Discussion

In the Introduction section, the authors described A. absinthium has pharmaceutical and 48 medicinal effects such as antimicrobial [9], insecticidal [10], antiparasitic [11], antitumor [12], 49 antipyretic [13], hepatoprotective [14,15] and antioxidant activities [16,17]. As you described in the introduction section, A. absinthium has an antioxidant activities. In this manuscript, however, you showed that these extract promoted ROS production in BEL-7404 cells and H22 cells. The authors should address this seemingly paradoxical consequences in the revised manuscript.

Answer: Thanks for the reviewer’s suggestion. We have discussed this point in the revised manusript.

‘We observed that the ROS levels in hepatoma cells treated with AAEE and AAEE-Pe were lower than that of untreated cells at the beginning of 3 h, although they were significantly upregulated after 6 h, suggesting that the extracts might have antioxidant activities. This is similar with other studies [16,17]. The increased ROS might be due to the reduction of Δψm induced by AAEE and AAEE-Pe treatment.’ 

Minor point

What kind of post hoc tests are you using in this study?

Answer: The post hoc multiple comparison is Tukey's Multiple Comparison Test.

We have added it in the revised manuscript.

 Please add the representative bands of Internal Standard in fig2C.

Answer: Thanks for the reviewer’s suggestion. We have added β-actin as the internal standard in the revised manuscript.

Reviewer 2 Report

Wei and colleagues have prepared extracts from Artemisia absinthum and have tested their anti-tumorigenic potential in hepatocellular carcinoma cells. They report that artemisia extracts strongly suppress tumor cell growth in vitro and in vivo through induction of apoptosis and show evidence for an apoptotic mechanism involving endoplasmic reticulum stress and a mitochondrial-dependent pathway. The data are interesting and if the drugs can really be applied to patients with hepatic carcinoma, artemisia compounds may be a promising alternate drug treatment option for affected patients.

Major comments:

1) Was the artemisia powder used for extraction derived from specific parts of the plants (flower, leaves, seeds, etc.). What was the rationale to prepare 3 types (ethanol extract, petrol ether extract and ethyl acetate extract) of extracts from the powder.

2) In fig.1 it is obvious, that in particular in BEL cells maximum inhibitory effects are achieved after 24 h and subsequent treatment with extracts up to 72 h do not further inhibit cell growth. How can this be explained.

3) The statement that the extracts have no cytotoxic effects in normal liver cells is very optimistic as for instance AAEE-Pe has a stronger growth suppressive activity in normal liver cells than in BEL hepatoma cells (Fig. 1B vs. 1C).

4) In fig 4B the quality of the blots is poor, for instance the C-PARP band seems to be cut away in part for H22 cells and for the same cell, the C-caspase3 band is hardly visible. Moreover there is obviously no change in C-caspase-9 expression in BEl-cells after extract treatment. Blots should either be improved or replaced by densitometric data. 

5) Have artemisia extracts already been used in humans and is anything known about the bioavailability and stability of active artemisia compounds.

Author Response

Major comments:

1) Was the artemisia powder used for extraction derived from specific parts of the plants (flower, leaves, seeds, etc.). What was the rationale to prepare 3 types (ethanol extract, petrol ether extract and ethyl acetate extract) of extracts from the powder.

Answer: We are sorry for missing information. The powder was made using the aerial parts of A. absinthium including stems, leaves, flowers and seeds. We have added it in the revised manuscript.

In this study, A. absinthium ethanol extract (AAEE) was firstly prepared, and then the subfractions of AAEE were prepared by petroleum ether and ethyl acetate according to the polarity of the solvent. The active components will distribute in different solvent. Petroleum ether may enrich oils, waxes, chlorophyll, volatile oils, free steroids and triterpenoids. Ethyl acetate may enrich alkaloids, organic acids, flavonoids and coumarin aglycons. This study provides the guides for the further isolation antitumor components.

2) In fig.1 it is obvious, that in particular in BEL cells maximum inhibitory effects are achieved after 24 h and subsequent treatment with extracts up to 72 h do not further inhibit cell growth. How can this be explained.

Answer: The possible reason is that BEL7404 cells are not very sensentive to these extracts. The low concentrations of these extract showed weak cytotoxic for BEL7404 cells even though treated longer time. But the high concentration of these extracts showed the time-dependent manner.   

3) The statement that the extracts have no cytotoxic effects in normal liver cells is very optimistic as for instance AAEE-Pe has a stronger growth suppressive activity in normal liver cells than in BEL hepatoma cells (Fig. 1B vs. 1C).

Answer: 25 μg/ml of AAEE-Pe showed similar inhibitory effect on BEL7404 and 

NCTC1469 cells but 75 and 150 μg/ml of AAEE-Pe showed higher cytotoxicity for BEL7404 cells than NCTC1469 cells, suggesting that BEL7404 cells are more sensitive to AAEE-Pe.

4) In fig 4B the quality of the blots is poor, for instance the C-PARP band seems to be cut away in part for H22 cells and for the same cell, the C-caspase3 band is hardly visible. Moreover there is obviously no change in C-caspase-9 expression in BEl-cells after extract treatment. Blots should either be improved or replaced by densitometric data.

Answer: Thanks for the reviewer’s suggestion. We have provided the new blots in the revised manuscript. Although the level of C-caspase-9 did not significantly changed, the level of caspase-9 was significantly reduced in BEL7404 cells, which is similar with H22 cells.

5) Have artemisia extracts already been used in humans and is anything known about the bioavailability and stability of active artemisia compounds.

Answer: A kind of liniment made with A. absinthium and several other herbs has been used to treat chronic eczema in China (Hongjun Zhu, Clinical Journal of Chinese Medicine, 2015, 18:65-66). A kind of lotion made with A. absinthium has been used to treat inflammation induced by pathogens in China and showed good stability (Wei Li and Haiying Zhang, Endemic Diseases Bulletin, 2010, 3:77-78). However, A. absinthium has not been used to treat cancers.

Round  2

Reviewer 1 Report

The manuscript revised by Wei et al has been well-corrected according to the comments from the reviewers. However, there is a concern about the resolution of all Figure. Especially, the quality of photomicrograph in Fig. 1 is poor. The author should be check the original image to a better image in the revised manuscript.

Author Response

The manuscript revised by Wei et al has been well-corrected according to the comments from the reviewers. However, there is a concern about the resolution of all Figure. Especially, the quality of photomicrograph in Fig. 1 is poor. The author should be check the original image to a better image in the revised manuscript.

Answer: We appreciate the reviewer's comments. We have provided new figures with good quality.

Reviewer 2 Report

I have no further comments.

Author Response

We appreciate the reviewer's comments.